# A Crucial Role of Proteolysis in the Formation of Intracellular Dinitrosyl Iron Complexes

**DOI:** 10.3390/molecules29071630

**Published:** 2024-04-05

**Authors:** Karolina E. Wójciuk, Jarosław Sadło, Hanna Lewandowska, Kamil Brzóska, Marcin Kruszewski

**Affiliations:** 1Nuclear Facilities Operations Department, National Centre for Nuclear Research (NCBJ), 05-400 Otwock, Poland; 2Centre for Radiobiology and Biological Dosimetry, Institute of Nuclear Chemistry and Technology, Dorodna 16, 03-195 Warsaw, Poland; h.lewandowska@ichtj.waw.pl (H.L.); k.brzoska@ichtj.waw.pl (K.B.); m.kruszewski@ichtj.waw.pl (M.K.); 3Centre for Radiation Chemistry and Technology, Institute of Nuclear Chemistry and Technology, Dorodna 16, 03-195 Warsaw, Poland; j.sadlo@ichtj.waw.pl; 4School of Health & Medical Sciences, University of Economics and Human Sciences in Warsaw, 59 Okopowa St., 01-043 Warsaw, Poland; 5Department of Molecular Biology and Translational Research, Institute of Rural Health, Jaczewskiego 2, 20-090 Lublin, Poland

**Keywords:** K562 cells, electron paramagnetic resonance, nitric oxide, reactive nitrogen species, dinitrosyl iron complexes, chelatable iron, glutathione

## Abstract

Dinitrosyl iron complexes (DNICs) stabilize nitric oxide in cells and tissues and constitute an important form of its storage and transportation. DNICs may comprise low-molecular-weight ligands, e.g., thiols, imidazole groups in chemical compounds with low molecular weight (LMWDNICs), or high-molecular-weight ligands, e.g., peptides or proteins (HMWDNICs). The aim of this study was to investigate the role of low- and high-molecular-weight ligands in DNIC formation. Lysosomal and proteasomal proteolysis was inhibited by specific inhibitors. Experiments were conducted on human erythroid K562 cells and on K562 cells overexpressing a heavy chain of ferritin. Cell cultures were treated with •NO donor. DNIC formation was monitored by electron paramagnetic resonance. Pretreatment of cells with proteolysis inhibitors diminished the intensity and changed the shape of the DNIC-specific EPR signal in a treatment time-dependent manner. The level of DNIC formation was significantly influenced by the presence of protein degradation products. Interestingly, formation of HMWDNICs depended on the availability of LMWDNICs. The extent of glutathione involvement in the in vivo formation of DNICs is minor yet noticeable, aligning with our prior research findings.

## 1. Introduction

An important reaction involving nitric oxide (^•^NO) within living cells is its interaction with iron and specific biological ligands, including proteins, peptides, and amino acids [1,2,3,4]. The interaction results in the formation of dinitrosyl iron complexes (DNICs) characterized by electron paramagnetic resonance (EPR)-detectable g_┴_ = 2.04 and g_║_ = 2.014 signals (the so-called 2.03 signals, in accordance with the average value of its g-factor [1,5]). DNIC formation was observed in cells and tissues exposed to endogenous [6] or exogenous ^•^NO [7]. The nature of biological ligands forming DNICs and their interactions are not well understood. EPR analysis of frozen solutions revealed that the anisotropy of g values of formed DNICs displays significant variations depending on the geometry, electronic structure and molecular weight of the involved ligands, and is an important indicator of ligand type and complex structure. Nitrite and ^•^NO can react with heme and non-heme iron proteins to form DNICs; however, the kinetics of in vivo formation and distinctions between DNICs derived from nitrite or ^•^NO remains unclear [8]. During the formation of DNICs, a chelatable iron might be also sequestered, resulting in an iron-starved phenotype. Thus, DNIC formation affects cellular iron homeostasis and alters the activity of iron-containing proteins, but also reduces the pro-oxidant capacity of cells [6].

DNIC formation may involve a variety of iron-containing proteins, such as those comprising iron-sulfur centers [9,10,11], heme groups [12,13] or non-heme iron [11,14,15]. Some examples of proteins that have been shown to form DNIC include mitochondrial aconitase [8,16,17,18], ribonucleotide reductase [19], cytochrome c oxidase [20] and nitrogenase [21]. However, the specific proteins that are targeted during DNIC formation can vary depending on the cell type and physiological context [8]. Biosynthesis of DNICs was first discovered in 1964 [22,23], and continued investigations revealed tetrahedral [(NO)_2_Fe(L)_2_] complex, as a natural and ubiquitous DNIC formed from interaction of ^•^NO with non-heme iron-sulfur ((Fe-S)) proteins and cellular labile iron pool [24,25,26,27,28,29,30,31,32,33,34].

DNICs’ involvement in various ^•^NO-mediated cellular and organismal functions has been described in several reviews [11,35]. There is an increasing awareness of the biological role of DNICs [11], and this warrants further studies on DNICs’ structure and on the pre-requirements for their biosynthesis. DNICs have been proposed as a “working form” of ^•^NO under physiological conditions [22]. On the other hand, high-molecular-weight DNICs (HMWDNICs) accumulate in lipoprotein aggregates that contain partially hydrophobic fragments, such as lipid bilayers. These complexes are highly stable and inert, as demonstrated by their resistance to decomposition in the presence of iron chelators. HMWDNICs seem to be the most abundant form of DNICs observed in vivo; however, this might be biased by the fact that their enhanced stability may contribute to their higher abundance and better detectability in cellular extracts after protein fractionation [36]. The function of low molecular weight DNICs (LMWDNICs) as ^•^NO donors has been observed: administration of LMWDNICs to experimental animals resulted in the formation of high molecular weight protein-bound DNICs (HMWDNICs) due to the transfer of Fe(NO)_2_ groups from added complexes to proteins [37,38].

The objective of this study was to investigate the contribution of low- (LMW) and high-molecular-weight (HMW) ligands in the formation of DNICs. Our previous study revealed that DNICs are formed primarily in the endosomal/lysosomal fraction and degradation of iron-containing metallo-proteins is crucial for its formation in vivo [39]. This prompted us to undertake a more detailed investigation on the role of protein degradation in DNIC formation, through the use of proteolysis inhibitors of different specificity.

In this study, K562 human erythroid precursor cells were treated with proteolysis inhibitors followed by treatment with an •NO donor and a DNIC-specific signal was recorded with regard to shape and intensity. In addition, as glutathione appears to be the most abundant low molecular weight thiol ligand, its role in DNIC formation was also investigated.

## 2. Results and Discussion

Cellular proteins that might serve as HMW-ligands are degraded, either in lysosomes, cytoplasmic organelles containing various types of proteases active at acidic pH, or targeted for degradation to proteasomes in a ubiquitin-dependent process, both processes easy to control using appropriate inhibitors. On the other hand, level of the main potential LMW peptide ligand, glutathione, can be also controlled by the use of a specific precursor (NAC) or inhibitor (BSO) of its synthesis. In other words, the use of specific proteolysis inhibitors allows for elucidation of the role of HMW-ligands, whereas controlling the glutathione level allows for elucidation of the role of LMW-ligands in DNIC formation. The study also aimed to understand the relationship between both types of protein ligands in DNIC formation in various cellular compartments.

To further elucidate the role of HMW-ligands in intracellular DNIC formation, their cellular content was increased by the use of ferritin-overexpressing cells. Ferritin is an iron storage protein known to promote DNIC formation [40]. Thus, we aimed to explore how overabundance of HMW protein, such as ferritin, may affect the formation and stability of DNICs.

### 2.1. The Effect of Inhibitors of Lysosomal Proteolysis on DNIC formation

To estimate whether proteins degraded in the lysosome pathway are a source of DNICs components, we used ammonium chloride, which effectively inhibits lysosomal proteases by alkalization of the lysosome interior. Pretreatment of K562 cells with 10 mM NH_4_Cl induced a time-dependent decrease in DNIC formation (Figure 1). The result shows that proteins degraded in the lysosomes are an important source of DNIC components.

Further, two other proteolysis inhibitors were applied: ALLM (*N*-acetyl-*L*-leucyl-*L*-leucyl-*L*-methioninal) [41] or leupeptin (*N*-acetyl-*L*-leucyl-*L*-leucyl-*L*-argininal). ALLM, binds the active site of cysteine proteases, mainly calpains and cathepsins, and forms a covalent bond with the catalytic cysteine residue, thereby blocking their activity [42]. Leupeptin is a reversible inhibitor of serine, threonine and cysteine proteases, including cathepsins and calpains. It inhibits protein degradation in lysosomes and cytosol, but it is more potent against lysosomal proteases. Leupeptin also binds the active site of proteases and forms a covalent bond with the catalytic residues, thereby blocking their activity [43]. It has been shown to inhibit autophagy by blocking the breakdown of cellular components in lysosomes [44]. Both inhibitors prevented the formation of DNICs in a similar way but, unlike in the NH_4_Cl treatment, the effect was not time-dependent, and full intensity of inhibition was already observed after 2 h of treatment (Figure 2 and Appendix A). To maintain experimental coherence, all inhibitors were added for the same time (2, 4 or 6 h), so it cannot be excluded that peptide inhibitors act much more quickly than NH_4_Cl, and time-dependency appeared also in their case, but before the first experimental point (2 h).

These three inhibitors block primarily the same proteolysis pathway, namely the lysosomal pathway, thus we expected a similar effect. The observed discrepancy is likely to be due to the different modes of action. Peptide inhibitors interact directly with the active centers of enzymes, whereas the action of NH_4_Cl is indirect, through the alkalization of the intra-lysosomal environment. Since lysosomal mechanisms maintaining acidic pH are very efficient [45], apparently it takes more time to inhibit lysosomal enzymes.

### 2.2. The Effect of Proteasome Inhibitors on DNIC Synthesis

To examine the effect of proteasomal proteolysis on DNIC formation, we inhibited proteasome with 20 µM lactacystin (LAC) [46] or 25 µM MG-132 [41,47,48]. LAC covalently binds the active site N-terminal threonine residue of proteasome β-subunit to form an intermediate species, clasto-lactacystin β-lactone. Consequently, it inhibits proteasome-dependent proteolysis at concentrations of 20 μM or higher [46]. MG-132 is a specific, potent, reversible and cell-permeable proteasome inhibitor that reduces degradation of ubiquitin-conjugated proteins by the 26S complex without affecting its ATPase and iso-peptidase activities [49,50,51]. This substrate-mimicking peptide acts primarily on the chymotrypsin-like site in the β subunit of proteasome. Both inhibitors strongly inhibited formation of DNIC in DEANO-treated K562 cells. Extremely low or no EPR signals were detected, indicating that the treatment with proteasome inhibitors entirely stopped DNIC biosynthesis (Figure 3 and Appendix A). This result shows that proteins degraded in proteasomes are a very important source of DNIC components.

### 2.3. Effect of the Total Proteolysis INHIBITION on DNIC Synthesis

Two inhibitors were applied that inhibit major types of cellular proteases, MG-101 and MG-262. MG-101 (*N*-acetyl-*L*-leucyl-*L*-leucyl-*L*-norleucinal; ALLN) is a cell-permeable inhibitor of calpain I, calpain II, cathepsin and cathepsin L. It inhibits neutral cysteine proteases and the proteasome [44,50]. MG-262 is a highly potent and selective cell-permeable inhibitor of proteasome [52,53,54], and probably lysosome proteolysis [48]. MG-262, a boronic peptide acid, is a potent proteasome inhibitor that selectively and reversibly inhibits the chymotryptic activity of the proteasome [52,53], while leupeptin inhibits serine, cysteine and threonine proteases but does not inhibit α-chymotrypsin or thrombin. Leupeptin is a competitive transition state inhibitor and its inhibition may be relieved by an excess of substrate [55,56]. The experiments revealed that total inhibition of the cellular proteolysis obtained with these inhibitors completely prevented DNIC formation. As both inhibitors gave identical results, only results for ALLN are shown (Figure 4 and Appendix A).

### 2.4. The Effect of Glutathione Concentration Modulation on DNIC Synthesis

One of the aims of our studies was to evaluate the role of LMW thiol ligands in DNIC formation. Hence, the contribution of glutathione (GSH) in DNIC formation in K562 cells was evaluated. The cellular GSH content was modulated by treatment with *N*-acetylcysteine (NAC), which stimulates GSH synthesis by maintaining a high free cysteine pool, or by treatment with *D*,*L*-buthionine [S,R]-sulfoximine (BSO), a specific inhibitor of γ-glutamylcysteine ligase, the first enzyme of glutathione synthesis pathway.

As shown in Figure 5, treatment with DEANO did not affect cellular glutathione level. However, 24 h treatment with NAC stimulated glutathione synthesis and significantly increased its level. roughly by 40%. In contrary, treatment with BSO, as expected, decreased the glutathione level, again roughly by 60%.

To examine the role of glutathione in DNIC formation, cell cultures pretreated with NAC or BSO were subsequently treated with DEANO, and intensity of EPR signal was compared with cells treated with DEANO alone. Despite big differences in glutathione level (Figure 5), the intensity of EPR signal in DEANO-treated cells did not differ substantially. neither in cells with increased glutathione level (NAC treated) nor in cells with decreased glutathione level (BSO treated), as compared with untreated cells. However, a statistically significant difference was revealed between NAC- and BSO-treated cells (Figure 6). As the same cell cultures were used to obtain the data presented in Figure 5 and Figure 6, these data suggest that glutathione had only a negligible role in the formation of intracellular DNICs. Hence, we conclude that the inhibition of proteolysis is the decisive factor limiting the production of DNICs.

### 2.5. Cellular Low and High Molecular Ligand Fractions

The role of low and high molecular ligands in DNIC formation in vivo was further examined by cell fractionation. K562 cell homogenate was treated with DEANO and fractionated. The high intensity EPR signal from DNICs was detected in the whole cell lysate and in the cytosol. The cytosol fraction was further fractionated by centrifugal filtration to obtain fraction with ligands < 100 kDa. As expected, centrifugation through the molecular membrane with a cut-off value of 100 kDa eliminated the majority of HMW thiols, with a negligible effect on glutathione content (Figure 7). With a molecular mass 0.3 kDa, the same level of glutathione should be present in all fractions.

The intensity of EPR signal dramatically dropped when HMW proteins were filtered out (Figure 8). The EPR signal intensity in <100 kDa fraction was less than 40% of this of unfiltered cytosol. All together, these results further support the crucial role of HMW ligands in formation of DNICs.

### 2.6. Characterization of EPR Signal of Protein-Bound DNICs

LMWDNICs, which are not bound to proteins, exhibit a symmetrical EPR signal at room temperature with a g-value of 2.03. Hyperfine Structure (HFS) analysis of the EPR signal of DNICs with cysteine showed that these complexes contain pairs of molecules of cysteine and NO (nitric oxide). The hyperfine structure (HFS) in the EPR signal at 273 K is a result of the interaction between unpaired electron and nitrogen nuclei of NO ligands and protons of cysteine [57]. The signal became narrower as the temperature increased, indicating that these DNICs are highly mobile at higher temperatures, leading to an averaging of the anisotropic properties of the signal [57]. The anisotropic shape of the EPR signal at 273K of DNICs often encountered in biological systems suggests that these DNICs are bound to proteins [57,58,59].

Prior investigations by Lee et al. [60] laid the groundwork for understanding of EPR signal symmetry in ferritin-bound DNICs. An illustration of how the EPR signal at 77K is affected by specific ligands involved in the formation of protein bound DNICs was provided, demonstrating that the EPR signal of ferritin-originated DNIC was a composite of signals from complexes bound either to histidine or cysteine residues of the protein. Notably, these two distinct species generated EPR signal that shared a common central value of approximately 2.03. However, they exhibited dissimilar symmetries, with the histidinyl complex displaying a rhombic symmetry, whereas the cysteinyl complex exhibited an axial symmetry [27,60,61].

It is also worth noting that the symmetry of the EPR signal can also be influenced by the structural configuration of the protein ligands, as exemplified by the EPR spectra of DNICs associated with different isoforms of glutathione S-transferase [62,63]. Thus, DNICs bound to proteins with only one thiol group, such as bovine or human serum albumin, exhibit lower (rhombic) symmetry, resulting in an EPR signal with three different g-factor values. This decrease in symmetry suggests that DNICs incorporate a ligand, likely to be a histidine residue of the protein. When these protein-bound DNICs come into contact with an excess of LMW thiols (e.g., cysteine or glutathione), they undergo a reversible transformation, replacing the protein’s histidine ligand with the LMW thiol. This change in coordination results in a more axially symmetrical EPR signal. In cells and tissues, it is possible that DNICs can contain either two or one HMW protein ligand. In the latter case, the LMW ligand is incorporated into the DNIC that produces an EPR signal with an axially symmetrical tensor of the g-factor (g⊥ = 2.04 and g║ = 2.014) [35].

In this work, FERH overexpression (K562/FERH cells) was used as a strategy to promote the formation of larger and more stable HMWDNICs in order to investigate their distinctive properties and characteristics. The EPR signals recorded in wild-type K562 cells or K562/FERH cells differed substantially (Figure 9). Comparison of spectra showed that, at 77 K, the EPR signal of DNICs in K562/FERH cells showed a rhombic symmetry with g-factors 2.05, 2.03 and 2.014. This symmetry was lower than the one observed in the wild-type K562 cells, in which the signal showed the axial symmetry. Interestingly, the shape and g-factor symmetry of EPR signals of wild type K562 cells in which the physiological proteolysis was inhibited by NH_4_CL treatment changed from axial to rhombic. The contribution of rhombic signal increased as inhibition progressed, while for untreated cells the axial signal predominated. After 4 h of incubation, the EPR signal showed rhombic symmetry with g-factors 2.05, 2.03 and 2.014. As stated in the literature, this type of spectral shape is associated with the presence in the complex of two different ligands: thiol and non-thiol, descended from one molecule [5,35]. The rhombic g-tensor symmetry of DNIC EPR spectrum is observed for HMWDNICs with proteins. This might suggest that HMWDNICs were visible in the cells after incubation with NH_4_Cl, while the LMWDNIC-associated signal predominated in cells before proteolysis inhibition. We speculate that stability of HMWDNICs is higher than that of LMWDNICs, but formation of LMWDNICs is necessary for synthesis of HMWDNICs. This would explain the decrease of signal intensity observed on Figure 10 and the absence of DNIC formation in cells in which the inhibition of proteolysis was complete (see Figure 3).

## 3. Materials and Methods

Unless otherwise indicated all chemicals were purchased in Sigma-Aldrich (Merck), Poznań, Poland.

### 3.1. Cell Lines and Chemicals

K562 human erythroid precursor cells (ATCC CCL-243) were grown in suspension in RPMI-1640 supplemented with 10% fetal bovine serum, 2 mM glutamine and antibiotics in humidified 5% CO_2_ incubator to a final density of 10^7^ cells per milliliter.

### 3.2. Preparation of HFER Transfected K562 Cells

RNA was isolated from K562 cells using RNeasy Mini Kit (Qiagen, Hilden, Germany) according to manufacturer’s protocol. Ferritin heavy chain mRNA was reverse transcribed to cDNA using ThermoScript RT-PCR System (Invitrogen, Thermo Fisher Scientific, Waltham, MA, USA) and specific primer (5′-GCTTTCATTATCACTGTCTCCCAG-3′). The resulting cDNA was used as a template in polymerase chain reaction (PCR) with forward primer (5′-ATAATAAGCTTAGTCGCCGCCATGACGAC-3′) engineered to have HindIII site and reverse primer (5′-GACAGATCCAGCTTTCATTAT-CACTGTCTC-3′) engineered to include BamHI site. PCR was performed in a 50 µL mixture containing 5 µL of 10× Pfx Amplification Buffer, 1.5 µL dNTPs mixture (10 mM each), 1 µL MgSO4 (50 mM), 1.5 µL of each specific primer (10 µM), 1 µL (2.5 U) of Platinum Pfx DNA Polymerase (Invitrogen), 2 µL of template and 36.5 µL of nuclease-free water. The PCR program involved initial denaturation at 94 °C for 3 min, followed by ten cycles of denaturation at 94 °C for 0.5 min, annealing at 56 °C for 2 min, extension at 68 °C for 45 s, and then 25 cycles of denaturation at 94 °C for 0.5 min, annealing at 62 °C for 0.5 min, extension at 68 °C for 45 s, followed by a final extension at 68 °C for 7 min. The resulting PCR product was analyzed by 1.5% (*w*/*v*) agarose gel electrophoresis and purified using QIAquick PCR Purification Kit (Qiagen, Hilden, Germany) according to manufacturer’s protocol. The product was then double digested with BamHI and HindIII restriction enzymes (Fermentas, Thermo Fisher Scientific, Waltham, MA, USA) and ligated using T4 DNA Ligase (Invitrogen, Thermo Fisher Scientific, Waltham, MA, USA) into the BamHI/HindIII double digested pEGFP-N2 vector (Clontech, Mountain View, CA, USA). *E. coli* strain HB101 was transformed with the ligation reaction and transformants were selected on LB-agar plates containing 30 µg/mL kanamycin. Single colonies were selected and the sequences of the plasmids isolated using QIAprep Spin Miniprep Kit (Qiagen, Hilden, Germany) were analyzed to verify the presence of the correct insert. Plasmid isolation, digestion with restriction enzymes and ligation were performed according to the manufacturer’s protocols. Constructed plasmid encoded ferritin heavy chain with EGFP fused to its C terminus. Plasmid coding FERH-EGFP and empty pEGFP-N2 plasmid (coding EGFP only) were transfected to K562 cells by electroporation using ECM 600 Electroporation System (BTX Instrument Division, Harvard Apparatus, Holliston, MA, USA) according to manufacturer’s protocol. After electroporation, cells were cultured in a medium containing 500 µg/mL G418 (Gibco, Thermo Fisher Scientific, Waltham, MA, USA). Individual clones were selected and stable expression of FERH-EGFP or EGFP was confirmed by fluorescent microscopy. The transfected cell line was designated as K562/FERH.

### 3.3. Inhibition of Proteolysis

The cells were treated with the following inhibitors: 15 mM of ammonium chloride (lysosome inhibitor; Sigma Aldrich, Steinheim, Germany), 100 μg/mL of leupeptin (lysosome inhibitor; Sigma, Steinheim, Germany), 50 μM ALLM (*N*-acetyl-*L*-leucyl-*L*-leucyl-*L*-methioninal; lysosome inhibitor; Sigma, Steinheim, Germany), 20 μM lactacystin (proteasome inhibitor; Sigma, Steinheim, Germany), 25 μM MG-132 (*Z*-Leu-Leu-Leu-al; proteasome inhibitor, Sigma, Steinheim, Germany), 50 μM MG-101 (ALLN, *N*-acetyl-*L*-leucyl-*L*-leucyl-*L*-norleucinal, a calpain and cathepsin inhibitor; Sigma, Steinheim, Germany) or 5 μM MG-262 (*Z*-Leu-Leu-Leu-B(OH)_2_; proteasome inhibitor; BioMol, Hamburg, Germany). NO donor was added to generate DNICs.

### 3.4. Generation of DNICs

After 2, 4 or 6 h of treatment with proteolysis inhibitor in the complete medium at 37 °C, the cells were treated with •NO donor (diethyl-ammonium (*Z*)-1-(*N*,*N*-diethylamino)diazen-1-ium-1,2-diolate, DEANO) for 15 min at 37 °C, at concentration 70 μM. The final ^•^NO concentration in the culture medium was 100 μM). The donor spontaneously dissociates in a pH dependent, first-order process with a half-life of 2 min at 37 °C, pH 7.4. Untreated cells served as a reference.

### 3.5. EPR Measurements

DNIC formation was monitored by electron paramagnetic resonance (EPR). Samples were prepared according to the following procedure: NO-treated cells were centrifuged for 10 min at 250× *g*, resuspended in 200 µL of PBS buffer and flash-frozen in liquid nitrogen. EPR spectra were measured on an X-band Bruker 300e spectrometer. All spectra were recorded at 77 K with microwave power of 1 mW and modulation amplitude of 3 G, time constant 41 ms. The characteristic EPR signals corresponding to DNIC formation were obtained. Once the EPR signal intensity had been measured, the protein concentration in each sample was assayed by the Bradford method [64]. The graphic below presents the treatment scheme for inhibitor-treated cells (Figure 11). 

### 3.6. Preparation of Subcellular Fractions

To investigate the role of ligands in DNIC formation in various cellular compartments, cell homogenates were treated with DEANO and fractionated by differential centrifugation. Approximately 10^8^ cells were washed with phosphate-buffered saline (PBS) and pelleted at 300× *g* for 10 min. The pellet was resuspended in the ice-cold buffer A (250 mM sucrose; 1 mM EDTA; 20 mM *N*-(2-hydroxyethyl)piperazine-*N*’-ethane-sulfonic acid) and broken in a Potter–Elvehjem homogenizer. The homogenate was centrifuged through the Centricon centrifugal filter devices (Amicon, Miami, FL, USA) with cut-off values of 100,000 MWCO (molecular weight of ligands < 100 kDa) and 30,000 MWCO (molecular weight of ligands < 30 kDa) at 1000× *g* for 10 min. EPR spectra were recorded for each fraction separately and recalculated for protein content.

### 3.7. Glutathione and Thiol Determination

Total content of compounds containing thiol groups was measured fluorometrically with the use of mono-bromobimane (MBB, Sigma Aldrich), a non-fluorescent bi-mane dye that adds a fluorescent tag when reacting with thiol groups (Ex/Em of thiol conjugate = 380/475 nm). The cellular GSH content was also measured fluorometrically, but with the use of mono-chlorobimane (MCB, Sigma Aldrich), a bimane dye, specific for glutathione, even in the presence of other thiol groups (Ex/Em of glutathione conjugate = 380/460 nm) [65].

In some experiments, to lower the cellular glutathione content, approximately 25 × 10^6^ of cells were incubated for 24 h at 37 °C in 25 mL complete medium containing 200 μM D,L-buthionine (S,R)-sulfoximine (BSO), a γ-glutamylcysteine synthetase inhibitor. In yet another series of experiments, to increase the cellular glutathione content, the cells were incubated with 10 mM N-acetylcysteine (NAC), a glutathione synthesis precursor, in the same conditions. After incubation, DEANO was added to generate DNIC, and the EPR signal was recoded and recalculated for protein content measured by the Bradford method.

### 3.8. Statistical Evaluation

The significance of differences between the mean values was analyzed by a pairwise comparison using Student’s *t*-test for independent samples.

## 4. Summary and Conclusions

The study provides significant insights into the formation of DNICs in vivo and the role of different ligands in this process. Inhibition of lysosomal proteolysis by ammonium chloride led to a time-dependent decrease in DNIC formation, indicating that proteins degraded in lysosomes are an important source of DNIC components. This was confirmed by the use of two different inhibitors of protein degradation in lysosomes and cytosol, namely ALLM and leupeptin. The treatment resulted in even stronger inhibition of DNIC formation. In line with this, inhibition of protein degradation in proteasome by LAC almost entirely stopped DNIC biosynthesis. Finally, total proteolysis inhibition using ALLN or MG-262, which inhibit various proteases, prevented DNIC formation entirely. Altogether, the presented results underscore the critical importance of protein degradation, both lysosomal and proteasomal, in DNIC biosynthesis.

Furthermore, the study revealed only limited contribution of glutathione (GSH), a low molecular weight thiol, in DNIC biosynthesis. Modulation of GSH level only slightly affected DNIC formation, indicating a minor role of GSH in cellular DNIC formation. Cell fractionation demonstrated that DNIC signal was more intense in high-molecular-weight fractions of cytosol, further supporting the role of larger ligands, such as proteins, in DNIC stabilization.

The study revealed also a distinct EPR signal with rhombic symmetry in ferritin-overexpressing cells, suggesting the formation of more stable DNICs with large ferritin molecules, incorporating two different ligands in the coordination zone. In contrast, in wild-type K562 cells, EPR signal was of an axial symmetry, implying a more symmetrical environment around the unpaired electron or, more likely, a faster rotation of a paramagnetic species due to the lower molecular mass of ligands. It is proposed that stability of HMWDNICs is higher than that of LMWDNICs, but formation of LMWDNICs is necessary for the synthesis of HMWDNICs.

In summary, this research provides valuable insights into the role of lysosomal and proteasomal proteolysis, as well as the impact of GSH levels and molecular ligands in the synthesis of DNICs. The study sheds light on the distinct effects of different proteolytic inhibitors and highlights the importance of larger ligands, especially proteins, in DNIC formation. Understanding these mechanisms is essential for comprehending cellular redox regulation and metal–ion homeostasis. The findings may have implications for developing novel therapeutic approaches targeting metal–ion metabolism and for understanding redox signaling in various pathologies related to oxidative stress and altered metal–ion balance.

### Future Directions

The effectiveness of protease inhibitors, such as ammonium chloride and leupeptin, in impeding lysosomal and proteasomal activities is well-documented [41,42,43]. Based on this well-established action of proteolysis inhibitors, we have made a preliminary attempt to examine the relationship between proteolysis kinetics and DNIC formation. In future iterations of our research, we plan to include direct evaluations of protease inhibitor effectiveness within our specific model as an additional aim. Our study provides a basis for a deeper exploration of the complex balance between proteolysis, ligand availability and DNIC formation in the cellular environment. The immediate direction of further research is to directly validate the efficacy of protease inhibitors in the specific context of DNIC formation.

Delving further into the mechanistic intricacies of proteolysis, an additional area for enhancement is the investigation of varied time intervals for the application of inhibitors, aimed at accurately pinpointing their kinetic impact on both proteolysis and DNIC formation. In present study, to maintain experimental coherence, all inhibitors were added for the same time (2, 4 or 6 h). Such kinetics were adopted due to our previous experience with proteolysis inhibition using ammonium chloride [39]. We decided on the treatment times having in mind the time necessary for proteolysis inhibitors to enter cells and inhibit proteolysis. A time of 1–2 h or more for treatment with NO donor is a standard time used in such experiments [64,65]. Yet, the results herein obtained indicate that peptide inhibitors act much More quickly than NH_4_Cl and time-dependency could be observed also in their case, but before the first experimental point (2 h). Investigating these inhibitors at shorter durations would be both interesting and valuable.

## Figures and Tables

**Figure 1 molecules-29-01630-f001:**
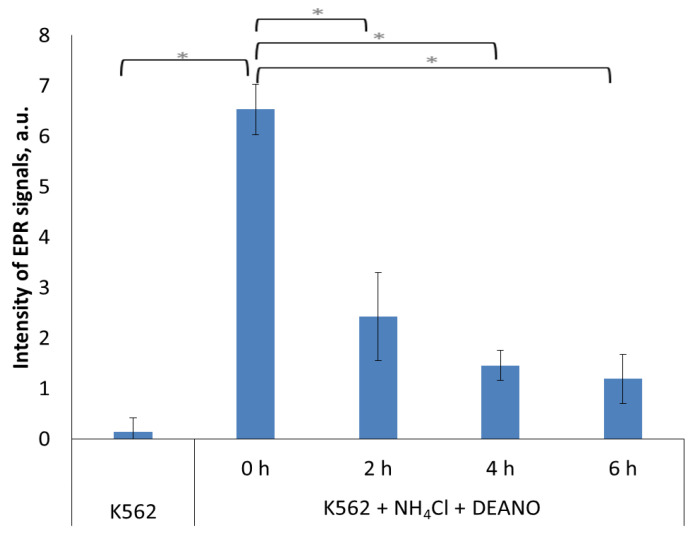
EPR signal induction in K562 cells incubated with 15 mM NH_4_Cl for indicated time (0, 2, 4, 6 h) and then treated with 70 μM DEANO for 15 min at 37 °C; mean ± SD, *n* = 3, * denotes statistically significant difference, *p* < 0.05.

**Figure 2 molecules-29-01630-f002:**
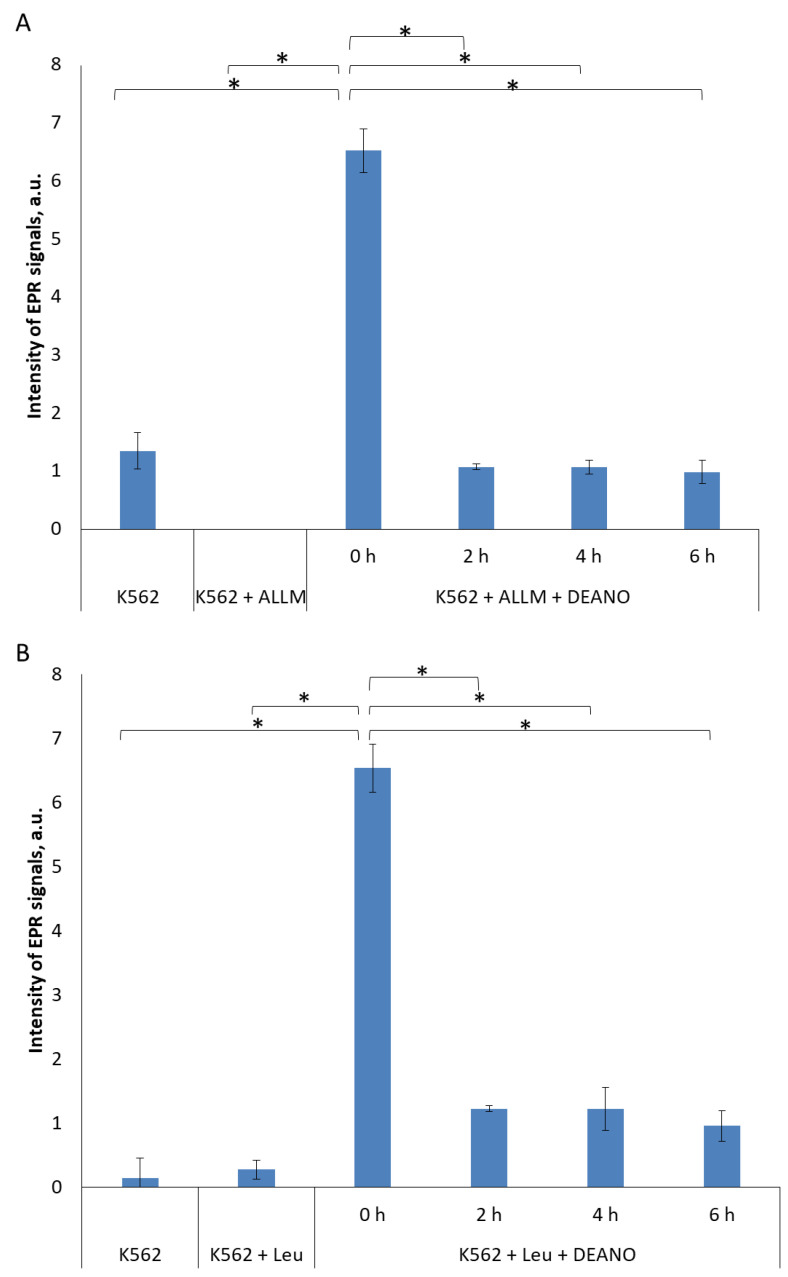
EPR signal induction in K562 cells incubated with 50 µM ALLM (**A**) or 100 µM leupeptin (**B**) for indicated time (0, 2, 4, 6 h) and then treated with 70 μM DEANO for 15 min at 37 °C; mean ± SD, *n* = 3, * denotes statistically significant difference, *p* < 0.05. EPR signal intensity for K562 + ALLM experimental point was below level of detection.

**Figure 3 molecules-29-01630-f003:**
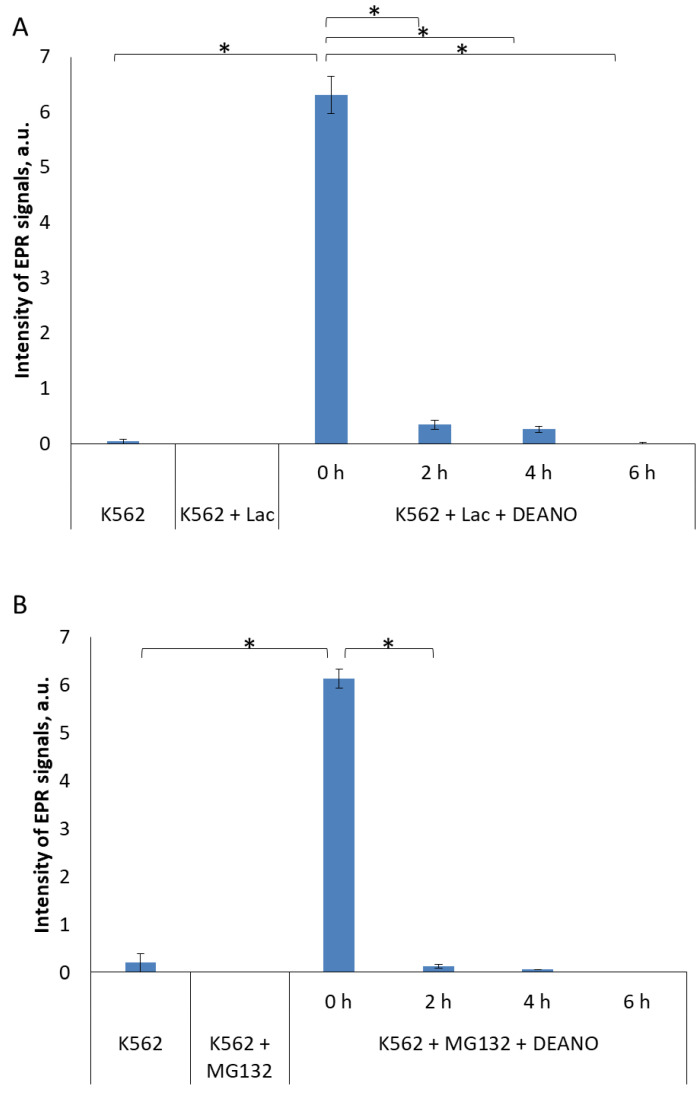
EPR signal induction in K562 cells incubated with 15 mM (**A**) 20µM lactacystin and (**B**) 25 µM MG132 for indicated time (0, 2, 4, 6 h) and then treated with 70 μM DEANO for 15 min at 37 °C; mean ± SD, *n* = 3, * denotes statistically significant difference, *p* < 0.05. EPR signal intensities for K562 + LAC and K562 + MG132 experimental points were below level of detection.

**Figure 4 molecules-29-01630-f004:**
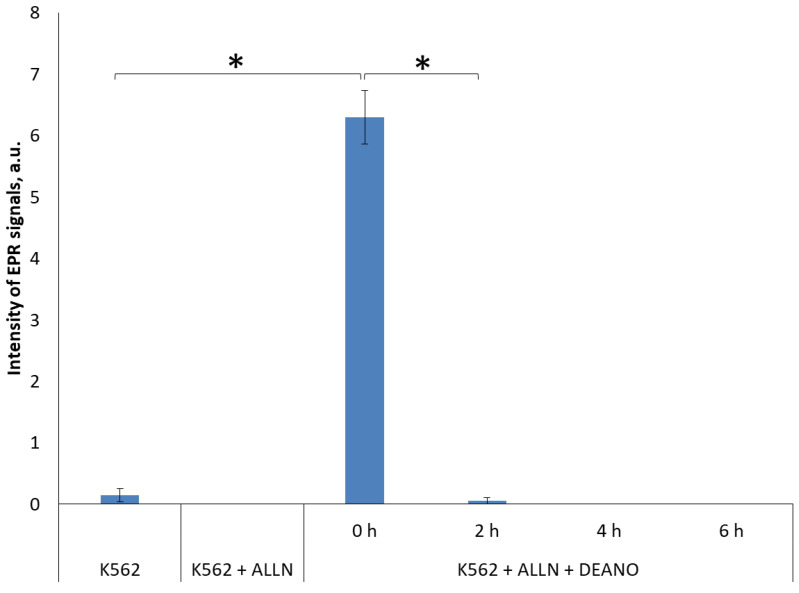
EPR signal induction in K562 cells incubated with 20µM ALLN for indicated time (0, 2, 4, 6 h) and then treated with 70 μM DEANO for 15 min at 37 °C; mean ± SD, *n* = 3, * denotes statistically significant difference, *p* < 0.05. EPR signal intensity for K562 + ALLN experimental point was below level of detection.

**Figure 5 molecules-29-01630-f005:**
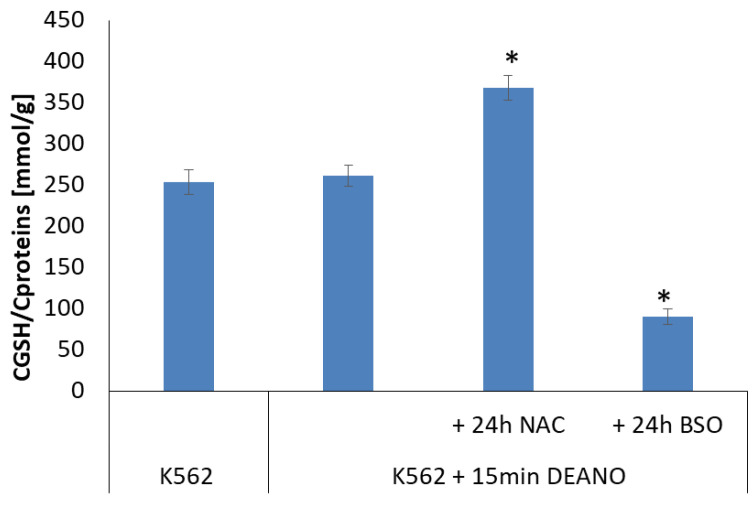
Cellular level of GSH in DEANO-treated cells previously treated with NAC or BSO. GSH levels were estimated with the monochlorobimane method; mean ± SD, *n* = 3; * denotes statistically significant difference versus untreated control cells, *p* < 0.05.

**Figure 6 molecules-29-01630-f006:**
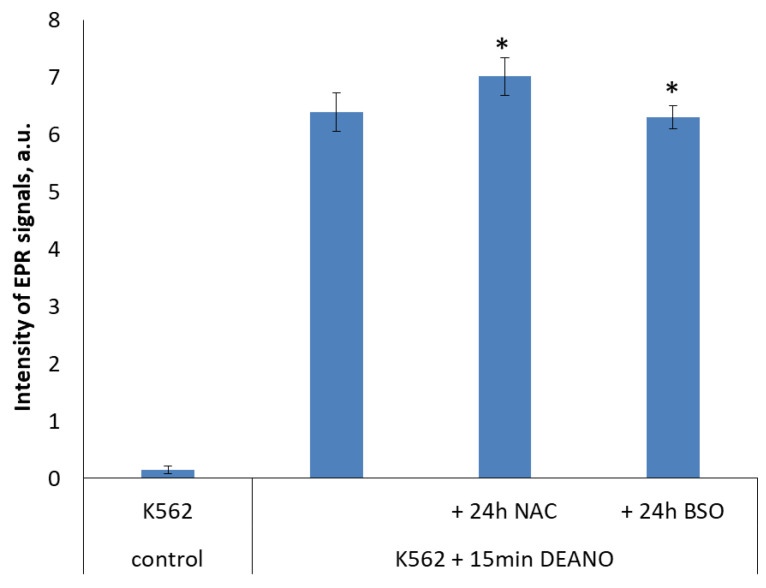
Intensity of EPR signal in cells differing in glutathione level. * denotes statistical significance between the marked levels, *p* < 0.05.

**Figure 7 molecules-29-01630-f007:**
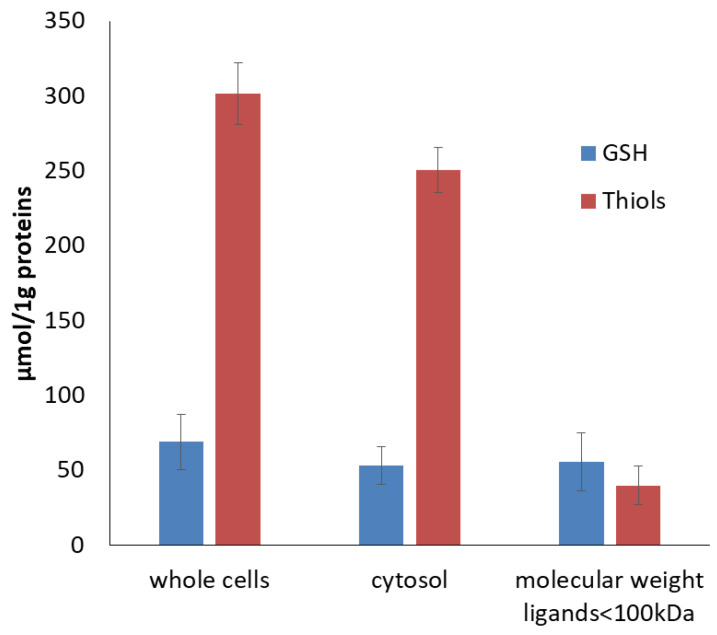
The levels of GSH and total thiols in cellular subfractions.

**Figure 8 molecules-29-01630-f008:**
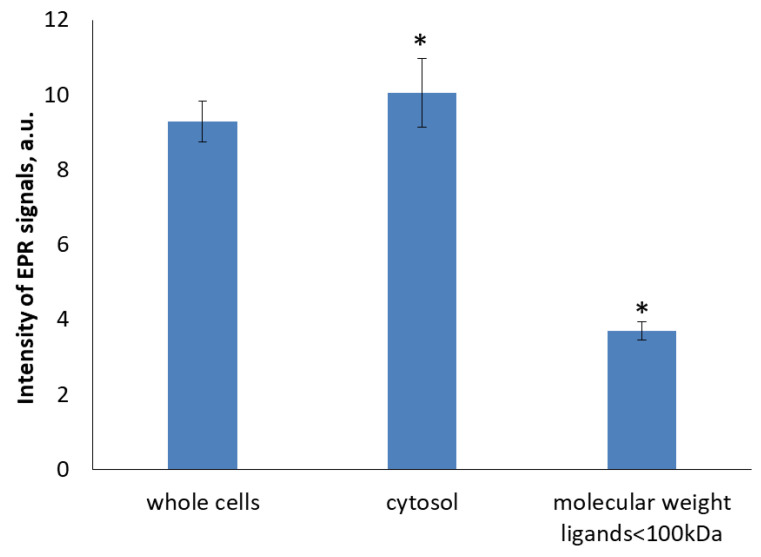
The intensity of the EPR signal in individual cell fractions. Cell homogenate was treated with 70 μM DEANO. * denotes statistically significant difference versus untreated control cells, *p* < 0.05.

**Figure 9 molecules-29-01630-f009:**
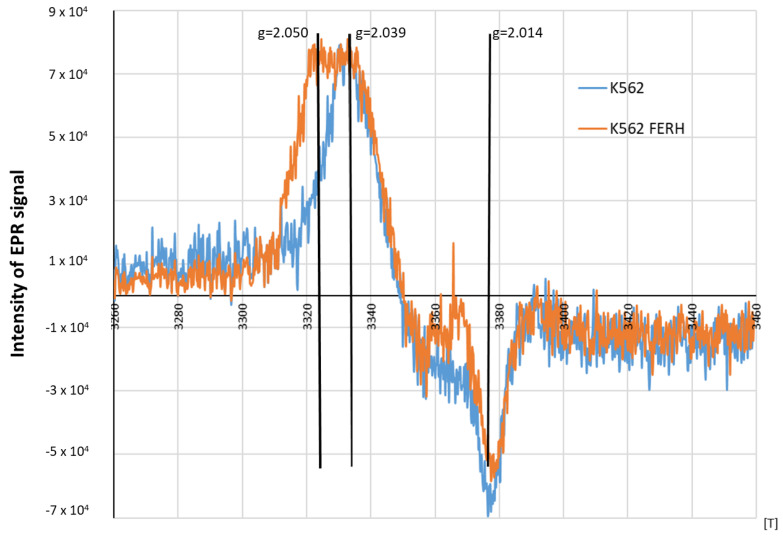
EPR spectra: K562 cells K562 cells overexpressing the heavy chain of ferritin (FERH). The cells were treated with 70 μM DEANO).

**Figure 10 molecules-29-01630-f010:**
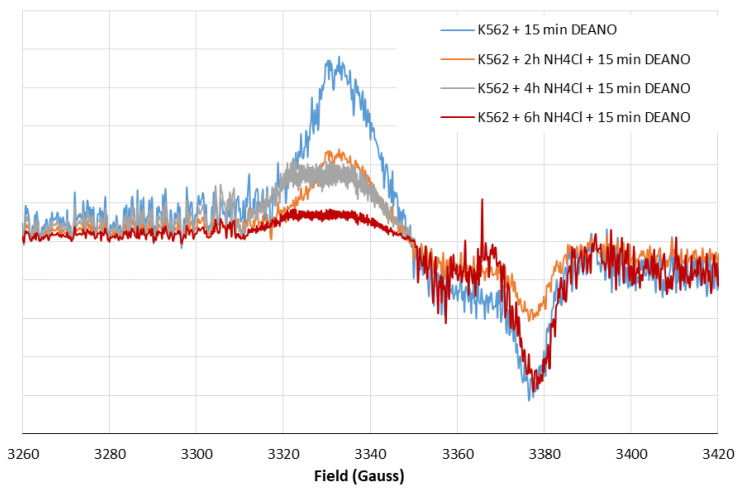
Shape of EPR spectra. Inhibition of EPR signal induction in K562 cells incubated with 10 mM NH_4_Cl for 2, 4, and 6 h and then treated with 70 μM DEANO for 15 min at 37 °C.

**Figure 11 molecules-29-01630-f011:**
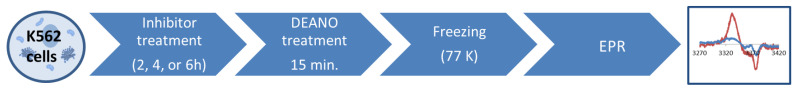
Infographic explanation of the treatment scheme. EPR spectrum from HMWDNIC (blue line) and LMWDNIC (red line).

## Data Availability

The raw data arising from this study are available from the corresponding author upon request.

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
