# Peer review of "A Crucial Role of Proteolysis in the Formation of Intracellular Dinitrosyl Iron Complexes"

_molecules, 2024, doi:10.3390/molecules29071630_

Round 1

Reviewer 1 Report

Comments and Suggestions for Authors

Wójciuk et al, have performed a systematic work and made a good attempt in the manuscript entitled "A crucial role of proteolysis in formation of intracellular di-2 nitrosyl iron complexes".

Strength of the manuscript: The manuscript presented is an extension of their work on the crucial role of lysosomal iron in forming dinitrosyl iron complexes. Their previous work demonstrates the formation of DNICs in the endosomal/lysosomal fraction. In the currently submitted work, the authors attempted to understand the function of low molecular weight DNIC as a nitric oxide radical donor. The approach they have taken is correct and reached good results.

weakness: Though the conclusion was well written by highlighting the observations of the experiments, the limitations of the work, along with the future studies, need to be presented.

The manuscript needs minor edits before it gets accepted

1. Title: article missing before formation. It should be  A crucial role of proteolysis in the formation of.....

2. In all the figures: A and B was mentioned in caps in figure but not in figure cation (it was mentioned as a and b). Need to be corrected.

3. units should be in correct format

4.  in vitro should be italics across the manuscript.

Reviewer 2 Report

Comments and Suggestions for Authors

Wójciuk et al. describe the role of proteolysis in the formation of dinitrosyl iron complexes, which serve as either 1) the working form of NO, or 2) the storage of NO. They hypothesize that inhibition of proteolysis will prevent formation of high molecular weight complexes, and therefore use a variety of protease inhibitors to assess the impact on dinitrosyl iron complex formation. They also modulate glutathione levels in order to assess the impact of this on the production of low molecular weight dinitrosyl iron complexes. The paper is generally well-written.

While the authors demonstrate the effectiveness of NAC and BSO on levels of GSH, they unfortunately do not demonstrate the effectiveness of the various protease inhibitors used on proteolysis. This is the greatest weakness I see in this paper, as the authors are making conclusions based on assumed effectiveness. (For example, lines 180-181 concludes that ‘this result shows that proteins degraded in the lysosomes are an important source of DNICs’; however, they have not shown that there is a change in lysosomal degradation.) I would recommend that they demonstrate that the protease inhibitors they use have the expected effects on proteolysis in the cells and conditions they use them.

Other comments are more minor:

1)     Lines 84-87; this paragraph seems misplaced, like an afterthought. It would fit better earlier in the introduction.

2)     Line 95; the authors state they have performed stable transfection, but do not describe how they selected for stable transfectants. Do they actually mean ‘transient transfection’?

3)     Figures (particularly figure 1) would be better without horizontal lines. They obscure the statistical markings, and make the figures generally cluttered.

4)     Line 199-200; ‘not time-dependent’.  Actually, it appears that the authors simply didn’t use a short enough time course. Why not add time points between 0 and 2 hrs?

5)     Line 234; ‘all types of cellular proteases’ is a big statement. I highly doubt that these two inhibitors inhibit all types of proteases; in fact, the authors description of these inhibitors immediately below suggests otherwise.

6)     Line 272, 292: neglectable = negligible.

7)     Lines 305-336 seem more like discussion, while the conclusion section is just a restatement of the results. I would recommend that they authors convert their ‘conclusions’ into a discussion section in which they consider their results in the context of the broader scientific literature.

Reviewer 3 Report

Comments and Suggestions for Authors

Please see the attached PDF file.

Comments on the Quality of English Language

Please check them over the text.

Round 2

Reviewer 2 Report

Comments and Suggestions for Authors

The authors have addressed all of this reviewer's concerns.  While I still feel that the experiments should have been designed with verification of the effectiveness of inhibitors in mind (even with well-validated reagents, strange things can happen), the authors otherwise present a well-written manuscript that effectively describes the impact of a number of chemical treatments on DNIC formation.

Reviewer 3 Report

Comments and Suggestions for Authors

Okay, the revised version is not sufficient for the publication.

Comments on the Quality of English Language

Please check the over all text before the next step.